## Research Article

depression; alcohol; health care system; global mental health; traditional healer

**Corresponding author:**
Tesfa Mekonen Yimer;
Email: t.yimer@uq.net.au

# Treatment-seeking behavior and barriers to mental health service utilization for depressive symptoms and hazardous drinking: The role of religious and traditional healers in mental healthcare of Northwest Ethiopia

Tesfa Mekonen Yimer[1,2,3] 🔵, Gary CK Chan[2], Habte Belete[1,2,3], Leanne Hides[1,2] and Janni Leung[1,2]

[1]School of Psychology, The University of Queensland, Brisbane, Australia; [2]National Centre for Youth Substance Use Research, The University of Queensland, Brisbane, Australia and [3]Psychiatry Department, Bahir Dar University, Bahir Dar, Ethiopia

## Abstract

Understanding mental healthcare seeking and associated factors is essential for planning mental health services. This study aimed to assess treatment seeking and barriers to care for depressive symptoms and hazardous drinking in a community sample of Northwest Ethiopia. A cross-sectional study was conducted to screen 1,728 participants for depressive symptoms ($n = 414$) and hazardous drinking ($n = 155$). Participants were asked whether they had sought mental healthcare. We also assessed the barriers to seeking mental healthcare. Logistic regression was used to identify associated factors. Among people with depressive symptoms, 14.3%, 15.5%, and 19.6% sought treatment from healthcare settings, non-healthcare settings, or any sources, respectively. Religious places (39.5%) were the most helpful treatment sources. People with low levels of internalized stigma (adj OR = 3.00 [1.41, 6.42]) and positive attitudes towards mental illness (adj OR = 2.84 [1.33, 6.07]) were nearly threefold more likely to seek depression treatment. No participants with hazardous drinking sought treatment from healthcare settings, and only 1.3% had sought help from families/friends. Over 97% of participants with depressive symptoms and hazardous drinking reported at least one barrier to treatment-seeking from a healthcare setting. Religious and traditional healers were as important as healthcare settings for treatment-seeking.

## Impact statement

This research provides valuable insights into treatment-seeking behavior and barriers to seeking treatment for depressive symptoms and hazardous drinking in a resource-limited setting. The findings highlight the significant treatment gap for individuals with depressive symptoms and hazardous drinking. This low treatment-seeking behavior emphasizes the urgent need to improve both the demand for mental health treatment and the availability of mental health services in the studied population. Furthermore, we identified important preferences and sources of treatment for depressive symptoms. Both healthcare and non-healthcare settings were reported as valuable sources of treatment, with religious places being particularly helpful for almost 40% of treatment seekers. These findings suggest that incorporating mental health services into existing religious and traditional pathways could enhance treatment accessibility and coverage. The study also sheds light on the barriers hindering treatment-seeking behavior. Attitudinal and stigma-related barriers were identified as the top obstacles to seeking treatment from healthcare settings. These barriers were even more prevalent than structural barriers such as limited resources and financial costs. This highlights the crucial role of addressing mental health literacy, reducing stigma, and increasing public awareness and acceptance of mental health services to promote treatment-seeking behavior. The implications of this study extend beyond the study population, as the findings are relevant to other low-income settings facing similar challenges in mental health services. By addressing the identified barriers and promoting a wider range of treatment options, including religious and traditional healers, mental health services coverage and utilization can be increased, reducing the treatment gap and improving the well-being of individuals with depressive symptoms and hazardous drinking.





## Background

Mental disorders are among the leading contributors of substantial health and economic burden worldwide (Arias et al., 2022; GBD 2019 Mental Disorders Collaborators, 2022). Pre-COVID-19

pandemic in 2019, one in eight (970 million) people worldwide were living with mental disorders (GBD 2019 Mental Disorders Collaborators, 2022). Depression was the second most prevalent mental disorder, causing the largest proportion (37.3%) of mental disorder disability-adjusted life years (DALYs), with a 25% worldwide increase due to the early COVID-19 pandemic (GBD 2019 Mental Disorders Collaborators, 2022; WHO, 2022a). Similarly, alcohol use disorder was the most prevalent substance use disorder in the world in 2019 (Castaldelli-Maia and Bhugra, 2022), resulting in 3 million deaths every year (WHO, 2019). In addition to the health burden, poor mental health is projected to cost the world economy approximately $6 trillion by 2030 (Bloom et al., 2012; The Lancet Global Health, 2020).

Lower- and middle-income countries (LIMCs) were estimated to bear 82% of people with mental disorders and 35% of the cost of poor mental health (Bloom et al., 2012; The Lancet Global Health, 2020; GBD 2019 Mental Disorders Collaborators, 2022; WHO, 2022b). For instance, Sub-Saharan Africa had the highest prevalence of depressive disorders in the global burden of disease estimate (GBD 2019 Mental Disorders Collaborators, 2022), and the economic loss due to mental disorders in East Africa reached 4% of the gross domestic product (Arias et al., 2022). In Ethiopia, mental illness, such as depression is the leading cause of years lived with disability (GBD 2019 Ethiopia Collaborators, 2022). The prevalence of depression ranged from 6.4% to 11.5% in Ethiopia (Bitew, 2014; Rathod et al., 2016). Hazardous alcohol use is also common in Ethiopia where the prevalence reached 8.9%–13.9% (Ayano et al., 2019; Zewdu et al., 2019).

Effective treatment options for mental illnesses exist. Pharmacological, psychosocial and community-based interventions are effective in treating mental disorders, including depression and alcohol use disorder in both high-income and low-income settings (Patel et al., 2007). Antidepressants (Cipriani et al., 2018), psychological interventions including cognitive behavioral therapy (Ijaz et al., 2018), and internet-based behavioral interventions (Karyotaki et al., 2021) or a combination of them (Malhi and Mann, 2018; Cuijpers et al., 2020) are effective in treating depression. A randomized controlled trial conducted in Zimbabwe demonstrated that problem-solving therapy, delivered by lay health workers, improved symptoms of common mental disorders (Chibanda et al., 2016). Likewise, a community-based rehabilitation intervention in Ethiopia, administered by lay health workers combined with facility-based care, has been demonstrated effective in reducing disability in patients with schizophrenia (Asher et al., 2022). Similarly, alcohol use disorder can be treated effectively using brief behavioral interventions in primary care, internet-based psychosocial interventions, psychological therapies, and pharmacological treatments (Connor et al., 2016). There is growing evidence in Sub-Sharan Africa that psychosocial interventions, including brief intervention and motivational interviewing, are promising in reducing heavy episodic drinking (Sileo et al., 2020) and promoting abstinence (Sileo et al., 2021). Increasing treatment coverage is one of the effective ways to minimize the health and economic burden of mental disorders.

Despite the considerable disease burden and available effective interventions, treatment-seeking is very low for mental disorders, particularly for depression and alcohol use disorder. The low treatment-seeking behavior creates a large gap between the prevalence and treatment of the illness. The prevalence of treatment-seeking is widely variable across countries, but only one in six people with alcohol use disorder (Mekonen et al., 2021a) and one-third of people with depression (Mekonen et al., 2021b) have received treatment globally. Limited evidence indicated that the treatment gap is even larger in LMICs where only 17% of people with depression and 9% of people with alcohol use disorder received treatment (Mekonen et al., 2021a,b). In Ethiopia, treatment-seeking was around 23% for depression (Hailemariam et al., 2012; Rathod et al., 2016) and 13% for alcohol use disorder (Rathod et al., 2016; Zewdu et al., 2019). The treatment rates for depression and alcohol use disorder are even lower than the treatment rate reported for psychotic disorders (40%) (Fekadu et al., 2019). Moreover, even this low rate of help-seeking typically occurs after a prolonged delay, with health facilities often being the last resort (Senait et al., 2020; Baheretibeb et al., 2021).

Multiple reasons have been identified why people with depression and alcohol use disorder do not utilize mental health services. Barriers to mental health service utilization can be attitudinal (mostly cultural and at the individual level) and structural (mostly at the health facility and policy level) (Bruwer et al., 2011; Andrade et al., 2014). Previous studies reported that attitudinal barriers were more critical in high-income settings and structural barriers were more likely in LMICs (Sareen et al., 2007; Roberts et al., 2018). However, available evidence suggested that both attitudinal and structural barriers are important in hindering or delaying treatment in low-resource settings (Andrade et al., 2014; Nadkarni et al., 2023). For example, a previous Ethiopian study reported that major barriers to seeking treatment for alcohol use disorder included low perceived need and a lack of knowledge about where to seek help (Zewdu et al., 2019). Similarly, another Ethiopian study reported that low perceived need and cost of treatment were the most common barriers to seek help for postpartum depression (Azale et al., 2016).

Understanding factors associated with seeking mental healthcare is essential for planning mental health service coverage and reaching untreated cases. Identifying the barriers that can hinder, delay, or discourage people from seeking mental health treatment is an important step towards improving mental health services and reducing the treatment gap. The overall aim of this study was to assess treatment-seeking behavior for depressive symptoms and hazardous drinking among a community sample in Merawi town, Northwest Ethiopia. The specific objectives were to (1) estimate treatment rates, (2) identify barriers to mental healthcare, (3) explore the treatment preferences and (4) assess the roles of religious and traditional healers.

## Methods

### Study setting and design

A cross-sectional study was conducted in Merawi town, the capital of Mecha District. Merawi town is in Northwest Ethiopia, 30 km from Bahir Dar City, the capital of Amhara Regional State. The town is organized with three kebeles (the smallest administrative unit in Ethiopia) with an estimated population of 42,159 in 2022 (ESS, 2022). One health center and one primary hospital serve the town and the surrounding rural population in the district.

### Participants

As part of a larger project, 1,785 participants were screened for depression and hazardous alcohol use. With a response rate of 97%, a total of 1,728 (52.4% females) adults aged 18 years and above were included in the screening process. Participants who were screened positive for depressive symptoms (*n* = 414) and/or hazardous drinking (*n* = 155) were used in the analysis of the current study which makes the final sample size 569.

## Sampling and data collection

We used a systematic random sampling method to select every other household in Merawi town. In each included household, one adult participant was selected randomly using a lottery method. The sample was allocated proportionally for each of the three kebeles. Data were collected in the private dwelling of the participants by a computer-assisted face-to-face interview in August 2021. Five psychiatry nurses, under the supervision of one author (H.B.) collected the data using a pre-loaded questionnaire on a tablet device. Participants with a positive screen for depressive symptoms or hazardous drinking were automatically prompted with additional questions related to treatment-seeking behavior and barriers to seek treatment from healthcare settings. When participants reported severe depressive symptoms, anxiety symptoms, and hazardous drinking, an automatic prompt instructs the data collector to refer them to the nearest health facility. Data collectors were trained on the procedures of data collection and materials using a training manual.

## Variables and measurement

### Socio-demographic variables

Participants were asked to report on their socio-demographic characteristics including gender, age, marital status, religion, education, income, and relative income. Relative income was assessed by asking participants to rate their income as poor, average, or well-off in relation to their neighborhood. This method of measuring relative income has been used in previous studies to assess how participants perceived their income compared to their neighbors (Fekadu et al., 2014; Mekonen et al., 2020).

### Psychosocial and related variables

Social support was assessed using the Oslo 3-item Social Support Scale (OSS-3). The OSS-3 is a widely used tool in Ethiopia (Fekadu et al., 2014; Zewdu et al., 2019; Mekonen et al., 2020) to assess the perceived number of close companions, level of concern from others, and perceived ease of getting help from families/neighbors/friends. A previous study (Kocalevent et al., 2018) reported good reliability for OSS-3 (Cronbach's alpha = 0.64) and it has acceptable internal consistency in the current study (Cronbach's alpha = 0.59). Social support was categorized as poor (OSS score 3–8), moderate (OSS 9–11), and strong (OSS 12–14) based on the sum score of OSS-3 (Bøen et al., 2012).

Perceived quality of life and perceived general health were assessed using single-item questions adapted from WHO's quality of life assessment tool (Skevington et al., 2004) and the Short Form-36 instrument (McHorney et al., 1993), respectively. The original scales for both the WHO's quality of life assessment tool and SF-36 are cross-culturally reliable assessments, with Cronbach's alpha values exceeding 0.82 for the specific domains that our single-item questions adapted (Brazier et al., 1992; Skevington et al., 2004). Participants were asked to rate their general health and quality of life as "very poor, poor, neither, good, very good." The items were re-coded as "poor, neither, good" during data processing.

Internalized stigma (due to depression and hazardous drinking) was assessed using a brief form of Internalized Stigma of Mental Illness Inventory (ISMI) (Boyd Ritsher et al., 2003) adapted in the Ethiopian context (Assefa et al., 2012). ISMI asked participants to rate their agreement (1 strongly disagree to 4 strongly agree) with statements related to self-stigma (e.g., "Having depression has

spoiled my life"). ISMI is a reliable measure in the Ethiopian context (Cronbach's alpha = 0.92) using its mean score as a cutoff point (Assefa et al., 2012; Zewdu et al., 2019). The Cronbach's alpha for ISMI was 0.78 in the current study. Attitude towards mental illness was assessed using six items adapted from the Reported and Intended Behavior Scale (RIBS) (Evans-Lacko et al., 2011), the Community Attitude towards Mental Illness Scale (CAMI) (Taylor and Dear, 1981), and the Mental Health Knowledge Schedule (Evans-Lacko et al., 2010) as relevant to the Ethiopian context. The items for the attitude towards mental illness include "I would be willing to work with someone with a mental health problem" (RIBS), "The mentally ill should be isolated from the rest of the community" (CAMI). The internal consistency for the attitude towards mental illness items was good in the current study (Cronbach's alpha = 0.70), with a scoring and categorization similar to ISMI.

### Mental health

The nine-item Patient Health Questionnaire (PHQ-9) was used to screen for depressive symptoms. PHQ-9 is validated in the Ethiopian context and has demonstrated good psychometric properties with 86% sensitivity and 67% specificity (Gelaye et al., 2013). Participants were asked to rate their experience of DSM-IV depressive symptoms in the past 2 weeks on a four-point scale (0 = not at all to 3 = nearly every day). The sum score for PHQ-9 ranged from 0 to 27, with higher scores showing severe depressive symptoms. PHQ-9 demonstrated high internal consistency in the current study (Cronbach's alpha = 0.85). Mild to severe depressive symptoms are indicated by a PHQ-9 score ≥5 (Kroenke et al., 2001). In addition to the PHQ-9, participants were also asked if they had experienced episodes of depressive symptoms in the last 12 months outside of the last 2 weeks (Luitel et al., 2017). Generalized anxiety disorder was assessed using the 7-item Generalized Anxiety Disorder scale (GAD-7) (Spitzer et al., 2006). GAD-7 has demonstrated high internal consistency in the current study (Cronbach's alpha = 0.83) and a total score of ≥10 is indicative of moderate to severe anxiety symptoms (Spitzer et al., 2006).

### Substance use

The 3-item Alcohol Use Disorders Identification Test–Consumption (AUDIT-C) was used to assess hazardous drinking. The AUDIT-C uses the first 3 items about alcohol consumption from the 10-item AUDIT and has comparable validity for detecting alcohol abuse or dependence at a cut-off point of 3 for females (99.3% sensitivity and 77.8% specificity) and 4 for males (88.4% sensitivity and 86.6% specificity) (Bush et al., 1998; Seth et al., 2015). AUDIT-C had high internal consistency in the current study (Cronbach's alpha 0.85). We also asked participants if they had ever used tobacco products and khat (Catha edulis, an evergreen stimulant leaf commonly used in East Africa and the Middle East).

### Treatment-seeking behavior

Treatment-seeking behavior was assessed by asking participants with depressive symptoms and/or hazardous drinking if they had ever sought treatment for their symptoms. Participants with an affirmative response for treatment-seeking were asked follow-up questions about the treatment source – healthcare settings (general medical care or specialized mental healthcare) or non-healthcare settings (religious places, traditional healers, sorcery, family/friend).

### Barriers to seeking professional help

Barriers to seeking professional help were assessed using the Barriers to Accessing Care Evaluation (BACE) scale (Clement et al., 2012). Participants were asked about a range of factors that hinder, delay, or discourage them from seeking or continuing professional help for depressive symptoms or hazardous drinking. The BACE scale has been modified to yes/no binary questions of 20 possible barriers and was previously used in the Ethiopian context (Azale et al., 2016; Zewdu et al., 2019). The BACE scale demonstrated good internal consistency in the current study (Cronbach's alpha 0.82 for hazardous drinking and 0.74 for depressive symptoms).

### Analysis

We described the participants' characteristics. The proportion of participants with depressive symptoms (PHQ-9 positive or last 12 months depressive symptoms) and hazardous drinking were reported. We also reported the proportion of participants who had ever sought treatment for depression and hazardous drinking.

For participants with depressive symptoms, treatment-seeking was cross-tabulated with participants' characteristics and other variables. Cross-tabulation and other further analyses were not conducted for participants who sought treatment for hazardous drinking because only two participants had reported treatment-seeking for their drinking problem. Proportions of barriers that hindered, delayed, or discouraged participants from seeking treatment for depressive symptoms and hazardous drinking were summarized.

Univariate logistic regression was used to estimate the unadjusted odds ratios of each variable in relation to treatment-seeking for depressive symptoms. Multivariable logistic regression was conducted to identify adjusted odds ratios for factors associated with treatment-seeking. Variables were entered into the final model using a backward elimination variable selection method. To adjust for the multiple comparisons, we used a 99.5% confidence interval and statistical significance was declared at a p-value less than 0.005 (Ioannidis, 2018). We use SPSS v.28 to perform the analysis.

## Results

### Participant characteristics

A total of 1,728 participants (52.4% female) were screened for depressive symptoms and hazardous drinking. The participants' age ranged from 18 to 90 years with a median age of 29 years. The majority of the participants were married (62.7%) and reported good perceived general health (84.8%). PHQ-9 depressive symptoms were 19.4% (95% CI: 17.5–21.2%). Self-reported depressive symptoms within the last 12 months were 13% (95% CI: 11.4–14.5%). Participants with either PHQ-9 or last 12 months depressive symptoms were 24% (95% CI: 21.9–26.0%). Only 3.2% (*n* = 56) reported both depressive symptoms and hazardous drinking. A significant difference in depressive symptoms was observed based on the participants' socio-demographic characteristics (age, marital status, and education), health-related (perceived general health and anxiety), and substance use (tobacco and khat) status (Table 1).

Hazardous drinking was reported by 155 (9% [95% CI: 7.6–10.3%]) of participants. Participants with hazardous drinking were mostly males (78.1%), married (65.8%), and in the age group of between 25 and 34 years (43.9%). There was a significant gender difference in hazardous alcohol use where hazardous drinking was 14.7% in males and only 3.8% in females. Tobacco and khat use

were also reported by 5.6% and 11.9% of participants, respectively. A significant difference in hazardous drinking was observed based on the participants' substance (tobacco and khat) use status (Table 1).

### Treatment-seeking behavior

Treatment-seeking behavior for hazardous drinking from any source was only 1.3%. No participant with hazardous drinking had sought help from healthcare settings. Only two participants (2/155) had sought help from their families/friends regarding their drinking problem.

Participants with depressive symptoms (PHQ-9 or last year depressive symptoms) (51.9% females) were mostly married (51.9%), in the age group of between 25 and 34 years (45.4%), maximum education of grade 12 (55.1%), and in the lower range of monthly income (58.7%). Among the 414 participants with depressive symptoms, 64.3% of them reported a high level of internalized stigma due to depression (Table 2).

Treatment-seeking behavior for depressive symptoms was 14.3% (95% CI: 10.9–17.6%) from healthcare settings, 15.5% (95% CI: 12.0–18.9%) from non-healthcare settings, and 19.6% (95% CI: 15.7–23.4%) from any treatment source (either healthcare or non-healthcare). Treatment-seeking behavior from any source of treatment was higher among participants with low levels of internalized stigma (29.7%) and participants with a positive attitude towards mental illness (28%). While the help-seeking behavior from most sources was relatively similar, specialized mental healthcare was the least preferred (3.1%) source of treatment. The low treatment-seeking behavior from a mental healthcare setting was more pronounced among the younger age group, those who reported strong social support, and participants with a higher level of internalized stigma due to depression (Table 2).

Almost 40% of treatment seekers for depressive symptoms reported religious places were helpful treatment sources. One-fifth of the treatment-seeking participants reported none of the treatment sources were useful in treating their depressive symptoms (Figure 1).

Unadjusted and adjusted estimates from logistic regression showed that attitudes about mental illness and internalized stigma of depression were important factors associated with treatment-seeking from any source of treatment. Participants with a positive attitude about mental illness (adjusted odds ratio (adj OR) = 2.84 [99.5% CI: 1.33–6.07]) and lower level of internalized stigma (adj OR = 3.00 [99.5% CI: 1.41–6.42]) were three times more likely to seek treatment. Other variables including sociodemographic, clinical, and psychosocial variables did not show a statistically significant association with treatment-seeking behavior in both unadjusted and adjusted logistic regression models (Table 3).

### Barriers to seeking treatment from healthcare settings

Almost all participants with depressive symptoms (98.6%) and hazardous drinking (97.4%) reported at least one barrier to seeking treatment from healthcare settings. Most of the participants (more than 73%) believed the problem would get better by itself or that they could handle it themselves. The median number of barriers to seeking treatment for depression and hazardous drinking was 4 (interquartile ranges = 3 for depression and 5 for hazardous drinking). Three-fifths of participants (62.6% for depression and 59.4% for hazardous drinking) reported at least four barriers. Ten

**Table 1.** Participant characteristics by depressive symptoms (PHQ-9 positive) and hazardous drinking status (*n* = 1,728)

| Variables | Total *n* (%) | Depressive symptoms (%) | $\chi^2$ test | Hazardous drinking (%) | $\chi^2$ test |
|---|---|---|---|---|---|
| Gender | | | | | |
| Male | 823 (47.6) | 19.9 | 0.588 | 14.7 | <0.001 |
| Female | 905 (52.4) | 18.9 | | 3.8 | |
| Age | | | | | |
| 24 and less | 395 (22.9) | 23.8 | 0.017 | 6.8 | 0.121 |
| 25 –34 | 771 (44.6) | 19.3 | | 8.8 | |
| 35 and above | 562 (32.5) | 16.4 | | 10.7 | |
| Marital status | | | | | |
| Married | 1083 (62.7) | 15.7 | <0.001 | 9.4 | 0.373 |
| Never married | 507 (29.3) | 27.6 | | 8.9 | |
| Divorced/widowed | 138 (8.0) | 18.1 | | 5.8 | |
| Education | | | | | |
| No formal education | 193 (11.2) | 14.0 | 0.028 | 7.8 | 0.642 |
| Grade 12 or less | 900 (52.1) | 21.6 | | 9.6 | |
| College or more | 635 (36.7) | 18.0 | | 8.5 | |
| Income | | | | | |
| <2,500 ETB | 978 (56.6) | 20.7 | 0.200 | 8.4 | 0.399 |
| 2,500–4,599 ETB | 377 (21.8) | 19.1 | | 8.8 | |
| 5,000 ETB and above | 373 (21.6) | 16.4 | | 10.7 | |
| Relative wealth | | | | | |
| Poor | 771 (44.6) | 19.8 | 0.169 | 8.7 | 0.904 |
| Average | 877 (50.8) | 19.7 | | 9.1 | |
| Well-off | 80 (4.6) | 11.3 | | 10.0 | |
| Perceived QOL | | | | | |
| Poor | 565 (32.7) | 20.7 | 0.619 | 9.4 | 0.154 |
| Neutral | 432 (25.0) | 18.5 | | 6.7 | |
| Good | 731 (42.3) | 18.9 | | 10.0 | |
| Perceived general health | | | | | |
| Poor | 97 (5.6) | 33.0 | <0.001 | 12.4 | 0.477 |
| Neither | 166 (9.6) | 33.1 | | 8.4 | |
| Good | 1465 (84.8) | 16.9 | | 8.8 | |
| Anxiety | | | | | |
| None/mild | 1642 (95.0) | 16.2 | <0.001 | 8.5 | 0.005 |
| Moderate to severe | 86 (5.0) | 80.2 | | 17.4 | |
| Social support | | | | | |
| Poor | 596 (34.5) | 25.5 | <0.001 | 10.4 | 0.231 |
| Moderate | 902 (52.2) | 16.9 | | 7.9 | |
| Strong | 230 (13.3) | 13.5 | | 9.6 | |
| Tobacco use | | | | | |
| Never | 1627 (94.2) | 18.7 | 0.003 | 7.9 | <0.001 |
| Yes | 101 (5.8) | 30.7 | | 26.7 | |
| Khat chewing | | | | | |
| Never | 1522 (88.1) | 18.3 | 0.003 | 8.0 | <0.001 |
| Yes | 206 (11.9) | 27.2 | | 16.5 | |

**Table 2.** Treatment-seeking for PHQ positive or last year depressive symptoms (*n* = 414)

| Variables | Total *n* (%) | Healthcare settings | | | Non-healthcare settings | | | Any treatment |
|---|---|---|---|---|---|---|---|---|
| | | General healthcare | Mental healthcare | Any healthcare | Religious places | Traditional healer and others[a] | Any non-healthcare | |
| Depressive symptoms | 414 (100) | 12.1 | 3.1 | 14.3 | 11.8 | 11.8 | 15.5 | 19.6 |
| Gender | | | | | | | | |
| Male | 199 (48.1) | 10.1 | 4.5 | 13.6 | 10.6 | 12.1 | 14.6 | 18.6 |
| Female | 215 (51.9) | 14.0 | 1.9 | 14.9 | 13.0 | 11.6 | 16.3 | 20.5 |
| Age | | | | | | | | |
| 24 and less | 112 (27.1) | 9.8 | 0.0 | 9.8 | 8.0 | 8.9 | 11.6 | 14.3 |
| 25–34 | 188 (45.4) | 14.4 | 4.8 | 17.6 | 15.4 | 14.9 | 19.1 | 24.5 |
| 35 and above | 114 (27.5) | 10.5 | 3.5 | 13.2 | 9.6 | 9.6 | 13.2 | 16.7 |
| Marital status | | | | | | | | |
| Married | 215 (51.9) | 14.9 | 4.2 | 17.7 | 14.0 | 13.5 | 17.7 | 22.8 |
| Never married | 167 (40.3) | 9.6 | 1.8 | 10.8 | 9.6 | 10.8 | 13.2 | 16.8 |
| Divorced/widowed | 32 (7.7) | 6.3 | 3.1 | 9.4 | 9.4 | 6.3 | 12.5 | 12.5 |
| Education | | | | | | | | |
| No education | 34 (8.2) | 11.8 | 2.9 | 14.7 | 11.8 | 2.9 | 11.8 | 14.7 |
| Grade 12 or less | 228 (55.1) | 11.4 | 3.1 | 13.6 | 11.4 | 11.4 | 14.5 | 19.3 |
| College or more | 152 (36.7) | 13.2 | 3.3 | 15.1 | 12.5 | 14.5 | 17.8 | 21.1 |
| Income | | | | | | | | |
| <2,500 ETB | 243 (58.7) | 10.7 | 2.5 | 12.8 | 10.7 | 10.7 | 14.0 | 18.9 |
| 2,500–4,599 ETB | 96 (23.2) | 13.5 | 4.2 | 15.6 | 11.5 | 11.5 | 14.6 | 18.8 |
| 5,000 ETB and above | 75 (18.1) | 14.7 | 4.0 | 17.3 | 16.0 | 16.0 | 21.3 | 22.7 |
| Perceived QOL | | | | | | | | |
| Poor | 139 (33.6) | 8.6 | 3.6 | 12.2 | 12.9 | 11.5 | 16.5 | 18.7 |
| Neutral | 105 (25.4) | 10.5 | 1.9 | 12.4 | 6.7 | 8.6 | 10.5 | 16.2 |
| Good | 170 (41.1) | 15.9 | 3.5 | 17.1 | 14.1 | 14.1 | 17.6 | 22.4 |
| Anxiety | | | | | | | | |
| None/mild | 345 (83.3) | 12.2 | 2.9 | 14.2 | 11.6 | 10.7 | 14.8 | 19.4 |
| Moderate to severe | 69 (16.7) | 11.6 | 4.3 | 14.5 | 13.0 | 17.4 | 18.8 | 20.3 |
| Social support | | | | | | | | |
| Poor | 175 (42.3) | 12.0 | 3.4 | 14.9 | 12.0 | 11.4 | 15.4 | 20.0 |
| Moderate | 195 (47.1) | 10.3 | 3.6 | 12.3 | 10.8 | 12.8 | 14.9 | 17.4 |
| Strong | 44 (10.6) | 20.5 | 0.0 | 20.5 | 15.9 | 9.1 | 18.2 | 27.3 |
| Tobacco use | | | | | | | | |
| No | 381 (92.0) | 12.9 | 3.1 | 15.0 | 12.1 | 12.6 | 16.0 | 20.5 |
| Yes | 33 (8.0) | 3.0 | 3.0 | 6.1 | 9.1 | 3.0 | 9.1 | 9.1 |
| Khat chewing | | | | | | | | |
| No | 352 (85.0) | 13.1 | 3.4 | 15.3 | 12.2 | 12.8 | 16.5 | 21.0 |
| Yes | 62 (15.0) | 6.5 | 1.6 | 8.1 | 9.7 | 6.5 | 9.7 | 11.3 |
| Hazardous drinking | | | | | | | | |
| No | 358 (86.5) | 13.1 | 3.1 | 15.1 | 12.3 | 11.5 | 15.4 | 19.6 |
| Yes | 56 (13.5) | 5.4 | 3.6 | 8.9 | 8.9 | 14.3 | 16.1 | 19.6 |

(*Continued*)

**Table 2.** (*Continued*)

| | | Treatment-seeking for depressive symptoms (%) | | | | | | |
| | | Healthcare settings | | | Non-healthcare settings | | | |
| Variables | Total n (%) | General healthcare | Mental healthcare | Any healthcare | Religious places | Traditional healer and others[a] | Any non-healthcare | Any treatment |
|---|---|---|---|---|---|---|---|---|
| Internalized stigma of depression | | | | | | | | |
| Low | 148 (35.7) | 16.2 | 6.8 | 21.6 | 20.3 | 20.9 | 26.4 | 29.7 |
| High | 266 (64.3) | 9.8 | 1.1 | 10.2 | 7.1 | 6.8 | 9.4 | 13.9 |
| Attitude about mental illness | | | | | | | | |
| Negative | 221 (53.4) | 8.6 | 2.3 | 10.4 | 6.8 | 5.9 | 9.0 | 12.2 |
| Positive | 193 (46.6) | 16.1 | 4.1 | 18.7 | 17.6 | 16.7 | 22.8 | 28.0 |

[a]Others = family/friends, wizard/sorcery; Any treatment = either healthcare or non-healthcare.

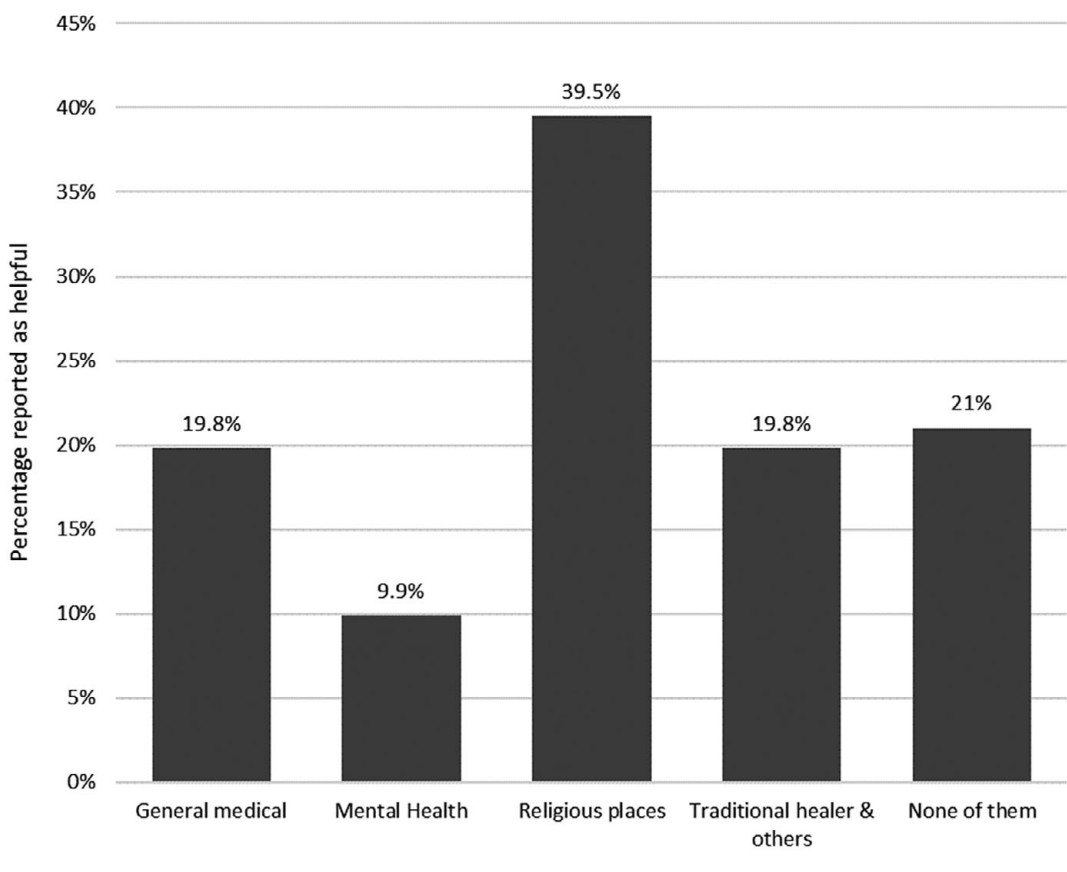

**Figure 1.** Treatment sources considered as helpful in treating depressive symptoms (*n* = 81).

or more barriers were reported by 9% of participants with depressive symptoms and 14.2% of participants with hazardous drinking.

The top reported barriers to treatment-seeking for both depression and hazardous drinking were attitudinal and stigma-related barriers. The top three barriers to seeking treatment for depression were participants wanting to solve the problem by themselves (75.1%), perceiving the problem would get better by itself (72.9%), and prefer to get alternative forms of care from traditional or religious healers (50.2%). Similarly, the top three barriers to treatment seeking for hazardous drinking were

participants' perceptions that their drinking problem would get better by itself (79.4%), wanting to solve their problem by themselves (78.1%), and perceiving that they do not have a drinking problem (56.8%).

A substantial proportion of participants also reported structural barriers. The most frequently reported structural barriers were the perception that "the process of seeking health services is complicated" (23.2% for depression and 20.6% for hazardous drinking) and "financial costs involved in treatment-seeking" (18.6% for depression and 18.1% for hazardous drinking) (Figure 2).

**Table 3.** Factor associated with treatment-seeking (from any sources) for depressive symptoms (*n* = 414)

| Variables | Unadjusted OR (99.5% CI) | Adjusted OR (99.5% CI) |
|---|---|---|
| Gender | | |
| Male | 1 | |
| Female | 1.13 (0.56, 2.26) | — |
| Age | | |
| 24 and under | 1 | 1 |
| 25–34 | 1.94 (0.79, 4.76) | 2.24 (0.87, 5.79) |
| 35 and above | 1.20 (0.43, 3.38) | 1.63 (0.54, 4.91) |
| Marital status | | |
| Married | 1 | |
| Never married | 0.68 (0.33, 1.43) | — |
| Divorced/widowed | 0.48 (0.10, 2.32) | |
| Education | | |
| No education | 1 | |
| Grade 12 or less | 1.39 (0.33, 5.85) | — |
| College or more | 1.55 (0.36, 6.72) | |
| Income | | |
| <2,500 ETB | 1 | |
| 2,500–4,599 ETB | 0.99 (0.42, 2.35) | — |
| 5,000 ETB and above | 1.26 (0.51, 3.10) | |
| Perceived QOL | | |
| Poor | 1 | |
| Neutral | 0.84 (0.32, 2.20) | — |
| Good | 1.25 (0.56, 2.78) | |
| Perceived general health | | |
| Poor | 1 | |
| Neither | 2.19 (0.39, 12.15) | — |
| Good | 1.69 (0.36, 7.97) | |
| Anxiety symptoms | | |
| None/mild | 1 | |
| Moderate to severe | 1.06 (0.42, 2.66) | — |
| Depressive symptoms | | |
| Mild/last year only | 1 | |
| Moderate to severe | 1.15 (0.54, 2.46) | — |
| Social Support | | |
| Poor | 1 | |
| Moderate | 0.84 (0.40, 1.79) | — |
| Strong | 1.50 (0.51, 4.45) | |
| Tobacco use | | |
| Never | 1 | |
| Yes | 0.39 (0.07, 2.21) | — |
| Khat chewing | | |
| Never | 1 | 1 |

*(Continued)*

**Table 3.** *(Continued)*

| Variables | Unadjusted OR (99.5% CI) | Adjusted OR (99.5% CI) |
|---|---|---|
| Yes | 0.49 (0.15, 1.56) | 0.40 (0.12, 1.37) |
| Hazardous drinking | | |
| No | 1 | |
| Yes | 1.01 (0.36, 2.78) | — |
| Attitude about mental illness | | |
| Negative | 1 | 1 |
| Positive | 2.79 (1.34, 5.80)* | 2.84 (1.33, 6.07)* |
| Internalized stigma of depression | | |
| Low | 2.62 (1.29, 5.32)* | 3.00 (1.41, 6.42)* |
| High | 1 | 1 |

— variables excluded in the adjusted model based a backward elimination variable selection method.
*\*p*-value <0.001.

## Discussion

We conducted a cross-sectional survey at Merawi town in Northwest Ethiopia to assess treatment-seeking behavior and identify barriers to seeking treatment for depressive symptoms and hazardous drinking. This study provided important implications in the mental health services of a resource-limited setting, highlighting the barriers in treatment-seeking and participants' preferences in the treatment sources.

Our finding showed that 19.6% of participants with depressive symptoms sought treatment either from healthcare or non-healthcare settings. Only 14% of participants sought treatment for depression from healthcare settings. Younger participants have reported relatively lower treatment rates for depression, while those in middle age have reported higher rates across all treatment sources. Although this difference is not significant in our regression model, previous studies have reported that middle-aged individuals are more likely to seek mental health treatment (Magaard et al., 2017; Roberts et al., 2018). Perceived stigma and preference for self-reliance may prevent young people from seeking help (Gulliver et al., 2010). No participant with hazardous drinking sought treatment from healthcare settings. Only two participants (1.3%) reported seeking help from their families/friends for their drinking problem, which is lower than the previous study (13% treatment-seeking) in the Southern part of Ethiopia (Zewdu et al., 2019). The relatively higher treatment-seeking in the previous Ethiopian study may indicate a better mental health awareness in the community due to the mental health programs implemented in that area (Lund et al., 2012). Our result is consistent with the previous meta-analyses that reported a range of 11–23% depression treatment rate (Mekonen et al. 2021b) and mostly 0% alcohol use disorder treatment rate (Mekonen et al., 2021a) in low-income countries.

Health workers could administer brief behavioral interventions (Pfeiffer et al., 2011), psychological interventions (Cuijpers et al., 2019), pharmacological treatments (Arroll et al., 2016; Connor et al., 2016), or a combination of them (Pampallona et al., 2004) to effectively treat depression and alcohol use disorder. However, 80% of participants with depressive symptoms and almost 100% of people with hazardous drinking in the current study had never

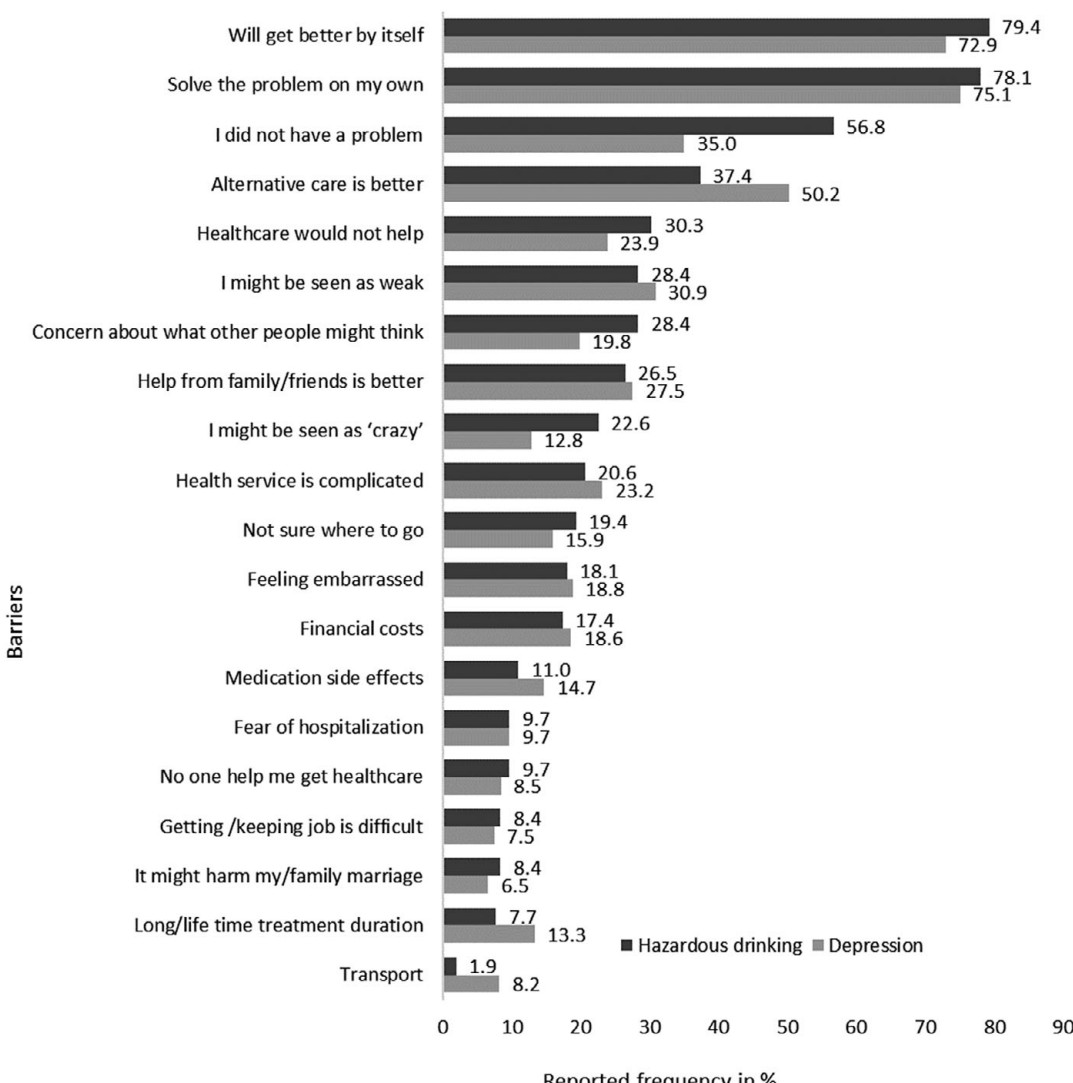

**Figure 2.** Barriers to seek treatment from healthcare settings for depression (*n* = 414) and hazardous drinking (*n* = 155).

sought help. These low rates of treatment-seeking created a large treatment gap, particularly for people with moderate to severe depressive symptoms who may benefit from pharmacological or behavioral interventions (NICE, 2022). Regardless of the treatment sources, 79% of treatment-seeking participants in the current study reported that their treatment-seeking helped to improve their depressive symptoms. Those groups of people who did not seek treatment could have benefited from the available helpful interventions.

Healthcare and non-healthcare settings were equally important sources of treatment for depressive symptoms. For instance, the prevalence of treatment-seeking was 12.1% from general medical care and 11.8% from religious and traditional healers. Interestingly, religious places were reported to be the most helpful source of treatment (39.5%) for depressive symptoms. A preference for religious and traditional healers over healthcare facilities was also reported in our previous qualitative study where seeking treatment from healthcare facilities was the last resort in the treatment-seeking pathway (Mekonen et al., 2022). Previous studies in Ethiopia reported that patients were comfortable taking treatments from healthcare facilities alongside religious/traditional healers

(Baheretibeb et al., 2021). As such, mental health services coverage could be increased by utilizing the existing help-seeking pathway of religious/traditional healers. Basic mental health training for religious leaders and traditional healers could also help increase referrals to health facilities (Mekonen et al., 2022).

Consistent with the previous studies from Ethiopia and other LMIC (Andersson et al., 2013; Zewdu et al., 2019), several barriers that hindered, delayed, or discouraged treatment-seeking from healthcare settings were reported by people with depressive symptoms and hazardous drinking. More than 97% of participants with depressive symptoms and hazardous drinking reported at least one barrier to seeking treatment from healthcare settings. According to the WHO Mental Health Atlas 2020 (WHO, 2021), the financial and human resources for mental health are meager in low-income countries with less than two mental health workers and mental hospital beds per 100,000 population. In Ethiopia, there were only 0.68 mental health workers per 100,000 population (WHO, 2021). With this paucity of mental health resources, one might expect structural barriers to be the most common barrier to seeking treatment from health facilities. However, consistent with the previous studies (Sareen et al., 2007; Andrade et al., 2014; Zewdu et al.,

2019), our study showed that the top reported barriers to treatment seeking from healthcare settings were attitudinal and stigma-related barriers.

It is worth noting that attitudinal and stigma-related barriers are not separate from structural barriers. Rather, structural barriers could contribute to the increased occurrence of attitudinal barriers. The assumptions related to the low perceived need for treatment (e.g., "the problem will get better by itself", "I do not have a problem", and "wanting to solve the problem by myself") can indicate the low level of mental health literacy. This low level of mental health literacy can be improved by addressing the structural barriers embedded at the health facility and policy levels (WHO, 2021). Without treatment, nearly 90% of people with depression will continue to have symptoms beyond 3 months (Mekonen et al., 2022).

In addition to the low mental health literacy, people may opt not to seek mental health services due to the stigma and discrimination attached to receiving mental illness diagnoses (Corrigan, 2004; Mekonen et al., 2022). This is also indicated in our study that seeking treatment for depressive symptoms was three-fold more likely in participants who had positive attitudes about mental illness (adj OR = 2.84) and participants who reported low internalized stigma of depression (adj OR = 3.00). Unless mental health literacy is improved, the public acceptance and perceived need for mental health services might be diminished (Jorm, 2000), and people with mental illness may avoid seeking mental health services (Mekonen et al., 2022). Population-based campaigns aiming in increasing mental health awareness and decreasing stigma could help in increasing mental health services coverage and utilization (Donovan et al., 2016; WHO, 2022b).

Consistent with the previous systematic review (Roberts et al., 2018), our study showed no difference in the sociodemographic characteristics of people who had sought treatment for depressive symptoms, compared to those who had not. This can be also another implication of attitudinal and stigma-related barriers being more critical at the individual level. Increasing mental health awareness, especially in the group of people who have a low perceived need to seek help could minimize the missed opportunity of accessing the available treatments.

Interpretation of the findings should consider the following limitations: first, the sample is from a semi-urban area in Northwest Ethiopia. The results may not be generalizable to the wider Ethiopian population. Descriptive statistics showed similar rates of depressive symptoms by gender, while in many contexts females generally have a higher rate. However, similar patterns of barriers to care were observed as indicated by the study conducted in the Ethiopian Southern part, especially for hazardous drinking (Zewdu et al., 2019). Second, the sample (cases with depressive symptoms and hazardous drinking) is relatively small which could be less powered in detecting cases and associated factors of treatment-seeking. Third, as we used screening tools to measure depression (PHQ-9) and hazardous drinking (AUDIT-C), the cases did not imply clinical diagnoses. While these tools are not diagnostic tools, they are widely used and have good psychometric properties including in the African contexts (Bush et al., 1998; Gelaye et al., 2013; Seth et al., 2015). Most of the measurement tools employed in this study (except PHQ-9 and ISMI) were not validated in the Ethiopian general population context, potentially introducing a bias. However, it is worth noting that these measurement tools have been widely used in previous Ethiopian studies and have demonstrated acceptable to high internal consistency in our study. Additional limitation to consider is that the assessment of

internalized stigma assumes that participants identified themselves as having depressive symptoms or hazardous drinking. However, this may not always be the case, introducing potential bias as some participants may not recognize these symptoms. Nonetheless, we took measures to mitigate this potential bias during the administration of internalized stigma assessment items by prompting participants to connect the ISMI questions with the symptoms they had previously reported.

## Conclusion

Only one in five people with depressive symptoms living in Merawi town had sought treatment from any source. Treatment-seeking for hazardous drinking was extremely rare where only 1.3% reported seeking help from family/friends. Both healthcare and non-healthcare settings were reported as valuable sources of treatment. A positive attitude about mental illness and low-level internalized stigma of depression was positively associated with seeking any treatment for depressive symptoms. The top reported barriers to treatment seeking from healthcare settings for both depressive symptoms and hazardous drinking were attitudinal and stigma-related barriers. The structural barriers including the financial cost of treatment-seeking and the complicated process of health services were also substantial.

**Open peer review.** To view the open peer review materials for this article, please visit http://doi.org/10.1017/gmh.2023.88.

**Data availability statement.** The data is available from the corresponding author on reasonable request.

**Acknowledgements.** We would like to acknowledge the study participants for their involvement in this research.

**Author contribution.** T.M.Y.: Conceptualization, design, protocol development, data analysis, synthesis, write-up, manuscript review and edits. G.C.K.C.: Conceptualization, synthesis, manuscript review and edits, supervision. H.B.: Data collection, manuscript review and edits. L.H.: synthesis, supervision, manuscript review and edits. J.L.: Conceptualization, protocol development, synthesis, supervision, manuscript review and edits. All authors read and approved the final manuscript.

**Financial support.** This research did not receive any specific grant from funding agencies. T.M.Y. is supported by The University of Queensland Research and Training Program (RTP) scholarship. G.C.K.C. and J.L. are supported by NHMRC Investigator grants. The data collection was financially supported by the College of Medicine and Health Sciences, Bahir Dar University.

**Competing interest.** The authors declare none.

**Ethics statement.** This study followed the ethical principles in accordance with the declaration of Helsinki. The University of Queensland (approval number: 2020/HE000945) and the College of Medicine and Health Sciences, Bahir Dar University (approval number: 009/2021) approved this study. A formal permission letter was obtained from the local administration to collect the data. Participants were informed about the confidentiality of their responses, and their right to refuse or to withdraw from the interview. Data collectors obtained informed consent from each participant. Participants with severe depressive symptoms, anxiety symptoms, and hazardous drinking were referred to the nearest health facility for the necessary treatment.

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
