## [Reviewer Report]

Dear Editor,

We are pleased to submit our manuscript entitled “Treatment-seeking behavior and barriers to mental health service utilization for depressive symptoms and hazardous drinking: The role of religious and traditional healers in mental healthcare of Northwest Ethiopia”, for consideration of publication at the Global Mental Health. 

This study aimed to assess treatment-seeking and barriers to seek treatment for depressive symptoms and hazardous drinking among community sample in Northwest Ethiopia. This study provided important implications in the mental health services of a resource limited setting, highlighting the barriers in treatment-seeking and participants’ preference in treatment sources. We believe the findings from this study can help to inform policymakers for planning mental health service coverage and reaching untreated cases. Identifying the barriers that can hinder, delay, or discourage people from seeking mental health treatment is an important step towards improving mental health services and reducing the treatment gap.

Only one in five people with depressive symptoms had sought treatment either from healthcare or non-healthcare settings. Treatment-seeking for hazardous drinking was extremely rare with only 1.3% had sought help from family/friends. Religious and traditional healers were equally important as healthcare settings in providing mental healthcare. Attitudinal and stigma-related barriers were the most frequently reported barriers.

This manuscript has not been previously published and is not under consideration in the same or substantially similar form in any other scientific journals. All authors listed have contributed sufficiently to the project to be included as authors, and all those who are qualified to be authors are listed in the author byline. All authors declare that they have no competing interests, and they also approved the manuscript as well as the submission of this paper. 

We appreciate your time, and we look forward to the response of the Editorial team to this manuscript.

Sincerely,

Corresponding author 

Tesfa Mekonen

---

## [Reviewer Report]

The authors present data on treatment seeking for depression and hazardous drinking in a cross-sectional sample of adults from a town in Northwest Ethiopia. While not exceptionally novel, the results add important data to the growing literature demonstrating increasing access to services is not enough to close the global mental health treatment gap. The sampling method provides a representative sample from this population, analyses are appropriate for the data, and results and conclusions clearly presented. My main question is why only depression and hazardous drinking were reported when data on anxiety was collected as well. Additionally, I have a few minor points that I believe will improve readability, interpretation, and impact of the manuscript.

General

Traditional places is perhaps too vague of a term that is difficult to interpret. From the text, it seems this refers to traditional healers, family/friends, and wizards/sorcery. I suggest the authors find another way to refer to this, either by saying something like “other informal care” or listing each when mentioned.

Methods

Why do the authors’ only report Cronbach’s alpha for some scales (ISMI, RIBS, CAMI, PHQ, GAD, AUDIT) in the study population, when it is unclear if the others (WHOQOL, OSS, SF 36, BACE) have good internal reliability in Ethiopian populations?

Were services offered to those who screened positive for depression, anxiety, or hazardous drinking?

Results/Discussion

Table 1 – It is interesting here that males and females have similar rates of depression symptoms, as in many contexts females have a significantly higher rate.

It is interesting, and perhaps surprising, that younger ages and those with more social support reported less treatment seeking for depression. The authors could expand on this in the discussion and compare to other studies. For example, one might think that younger people have less stigma toward MH and thus would have higher care seeking.

I think the authors should consider controlling for symptom level in the regression model, as the main “barrier to seeking care” is thinking one can get better on their own, and people with a 5-10 (mild depression) likely are not as concerned about formal care than those with higher/more severe symptoms. Similar issue with hazardous drinking, depending on severity a person may feel differently about their need for help.

---

## [Reviewer Report]

Review: Treatment-seeking behavior and barriers to mental health service utilization for depressive symptoms and hazardous drinking: The role of religious and traditional healers in mental healthcare of Northwest Ethio

General

The authors use different fonds, and the references section is poorly presented.

Abstract

Although the abstract is concise, the background does not provide the rationale for doing the study, the methods are clear, but results provide proportions without any odds and adjusted odds ratios.

Impact statement

Although the authors allude to the need for additional mental health care services, the absence of treatment seeking behaviour cannot be solved by increasing services but increasing the demand for services. This study did not explore the reasons for not seeking services.

Background

The rationale for the study is clear.

Methods

The methods are well presented. The choice of tools is also well presented and justified.

Analysis

The authors explain the data analysis process well.

Results

The presentation of the results is unusually dominated by proportions although the authors indicate that they used logistic regression in the analysis. the authors need to present the unadjusted and adjusted odds ratios. Even if the authors wished to present the proportions, they needed to present them with their 95% confidence intervals. This thus requires substantial revisions. I note the table 3 contains the odds ratios. Some important odds ratios were supposed to be extracted and presented in the results section but were not. This is an important omission.

Discussion

The discussion is appropriate but limited by the results as presented.

---

## [Reviewer Report]

This paper addresses the important topics of depression, anxiety and alcohol use in a randomly selected sub-urban population in Ethiopia. It focuses on prevalence, healthy seeking behavior and barriers. I have the following comments: - 

1. On introduction

(i) The first statement in the introduction is not referenced. It is also misleading. The authors should give an appropriate reference(s) or better still omit it. 

(ii) There is a need for more literature on Africa and more specifically for Ethiopia. Their emphasis on literature is not on Africa. 

(iii) There is a need to specify and detail which community-based interventions and which psychosocial interventions and also provide more evidence from Africa on the efficacy of these interventions. 

(iv) The lead author is quoting himself/herself inappropriately. Is this the work of Mekonen or was it referring to some original research which Mekonen had quoted in some of his publications? In that case, those original references are the ones to be quoted in this paper. 

(v) I do not think so – just to generalize that “There is lack of research”. There is a lot which can be obtained from literature search. Further, some of the references are too old to reflect current situation. 

2. Aims: These should be numbered – general and specific. This should be done at the end of the introduction. 

3. Methodology: The sampling procedure and the instruments are fairly well described and satisfactory. Although it is said that most of the instruments have been used in Ethiopia, it would greatly improve the paper to give the psychometric properties of those tools in general and in Ethiopia in particular. At the end of the discussion, the authors imply good psychometric properties for the tools they used without giving the actual properties except for one of them, although not adequate. These properties should be discussed in the methodology under special topic on tools/instruments. If that information on properties is not available, then this should be brought out by the authors as a major limitation of the study. 

4. Results: Well presented. A significant finding on the place of religion and traditional medicine on preferred source of care

---

## [Reviewer Report]

Dear Editor,

We are pleased to resubmit our revised manuscript entitled “Treatment-seeking behavior and barriers to mental health service utilization for depressive symptoms and hazardous drinking: The role of religious and traditional healers in mental healthcare of Northwest Ethiopia”, for consideration of publication at the Global Mental Health. Thank you for the comments and suggestions, which have improved the quality of our manuscript. We have been able to address each of the comments point-by-point.

This study aimed to assess treatment-seeking and barriers to seek treatment for depressive symptoms and hazardous drinking among community sample in Northwest Ethiopia. This study provided important implications in the mental health services of a resource limited setting, highlighting the barriers in treatment-seeking and participants’ preference in treatment sources. We believe the findings from this study can help to inform policymakers for planning mental health service coverage and reaching untreated cases. Identifying the barriers that can hinder, delay, or discourage people from seeking mental health treatment is an important step towards improving mental health services and reducing the treatment gap.

Only one in five people with depressive symptoms had sought treatment either from healthcare or non-healthcare settings. Treatment-seeking for hazardous drinking was extremely rare with only 1.3% had sought help from family/friends. Religious and traditional healers were equally important as healthcare settings in providing mental healthcare. Attitudinal and stigma-related barriers were the most frequently reported barriers.

This manuscript has not been previously published and is not under consideration in the same or substantially similar form in any other scientific journals. All authors listed have contributed sufficiently to the project to be included as authors, and all those who are qualified to be authors are listed in the author byline. All authors declare that they have no competing interests, and they also approved the manuscript as well as the submission of this paper. 

We appreciate your time, and we look forward to the response of the Editorial team.

Sincerely,

Corresponding author 

Tesfa Mekonen

---

## [Reviewer Report]

While the authors have attended to most of my concerns, the manuscripts has several typos and grammar to attend to. They need to closely attend to the language.

---

## [Reviewer Report]

The authors set out to determine the barriers to care for depression and alcohol use disorder. 

The authors should make reference to other workers who have done similar work in Ethiopia even if not completely identical. While it is okay to quote oneself, this should not be done to the exclusion of others. 

In addition to the reference on the Holy Water clinic, substantial work has been done in Ethiopia and in other countries in Africa on pathways to care. Therefore, for the paper to be more informative, a more inclusive review of literature would be appropriate. This would establish the gaps that their study intended to fill. 

There is need for revision in the discussion. While attempts are made to discuss the results, there is a mixture of discussion and what amounts to literature review. These references can be taken back to where they belong – literature review and then the authors could refer to what they found in the literature review.

---

## [Reviewer Report]

Dear Editor,

We are pleased to resubmit our revised manuscript entitled “Treatment-seeking behavior and barriers to mental health service utilization for depressive symptoms and hazardous drinking: The role of religious and traditional healers in mental healthcare of Northwest Ethiopia”, for consideration of publication at the Global Mental Health. Thank you for the comments and suggestions, which have improved the quality of our manuscript. We have been able to address each of the comments point-by-point.

This study aimed to assess treatment-seeking and barriers to seek treatment for depressive symptoms and hazardous drinking among community sample in Northwest Ethiopia. This study provided important implications in the mental health services of a resource limited setting, highlighting the barriers in treatment-seeking and participants’ preference in treatment sources. We believe the findings from this study can help to inform policymakers for planning mental health service coverage and reaching untreated cases. Identifying the barriers that can hinder, delay, or discourage people from seeking mental health treatment is an important step towards improving mental health services and reducing the treatment gap.

Only one in five people with depressive symptoms had sought treatment either from healthcare or non-healthcare settings. Treatment-seeking for hazardous drinking was extremely rare with only 1.3% had sought help from family/friends. Religious and traditional healers were equally important as healthcare settings in providing mental healthcare. Attitudinal and stigma-related barriers were the most frequently reported barriers.

This manuscript has not been previously published and is not under consideration in the same or substantially similar form in any other scientific journals. All authors listed have contributed sufficiently to the project to be included as authors, and all those who are qualified to be authors are listed in the author byline. All authors declare that they have no competing interests, and they also approved the manuscript as well as the submission of this paper. 

We appreciate your time, and we look forward to the response of the Editorial team.

Sincerely,

Corresponding author 

Tesfa Mekonen Yimer